# Discrimination of Head and Neck Squamous Cell Carcinoma Patients and Healthy Adults by 10-Color Flow Cytometry: Development of a Score Based on Leukocyte Subsets

**DOI:** 10.3390/cancers11060814

**Published:** 2019-06-12

**Authors:** Gunnar Wichmann, Clara Gaede, Susanne Melzer, Jozsef Bocsi, Sylvia Henger, Christoph Engel, Kerstin Wirkner, John Ross Wenning, Theresa Wald, Josefine Freitag, Maria Willner, Marlen Kolb, Susanne Wiegand, Markus Löffler, Andreas Dietz, Attila Tárnok

**Affiliations:** 1Clinic for Otorhinolaryngology, Head and Neck Surgery, University Hospital Leipzig, Liebigstr. 10-14, 04103 Leipzig, Germany; clara.gaede@googlemail.com (C.G.); j.wenning@gmx.net (J.R.W.); wald.theresa@web.de (T.W.); Josefine.Freitag@web.de (J.F.); maria.willner93@gmail.com (M.W.); marlen.kolb@medizin.uni-leipzig.de (M.K.); Susanne.Wiegand@medizin.uni-leipzig.de (S.W.); Andreas.Dietz@medizin.uni-leipzig.de (A.D.); 2LIFE-Leipzig Research Center for Civilization Diseases, University of Leipzig, Philipp-Rosenthal-Str. 27, 04103 Leipzig, Germany; susanne.melzer@zks.uni-leipzig.de (S.M.); jozsef.bocsi@gmx.de (J.B.); sylvia.henger@imise.uni-leipzig.de (S.H.); christoph.engel@imise.uni-leipzig.de (C.E.); Kerstin.Wirkner@medizin.uni-leipzig.de (K.W.); markus.loeffler@imise.uni-leipzig.de (M.L.); atarnok@hotmail.com (A.T.); 3Clinical Trial Centre Leipzig, University Leipzig, Härtelstr. 16-18, 04107 Leipzig, Germany; 4Institute for Medical Informatics, Statistics and Epidemiology (IMISE), University of Leipzig, Härtelstr. 10-18, 04107 Leipzig, Germany; 5Department of Precision Instruments, Tsinghua University, Beijing 100084, China; 6Fraunhofer Institute for Cell Therapy and Immunology IZI, Perlickstr. 1, 04103 Leipzig, Germany

**Keywords:** head and neck squamous cell carcinoma (HNSCC), multicolor flow cytometry, peripheral blood, leukocyte subsets, cancer biomarker, predictive value, monocytes-to-lymphocytes ratio (MLR), monocytes-to-granulocytes ratio (MGR), classical monocytes, propensity score matching

## Abstract

Background: Leukocytes in peripheral blood (PB) are prognostic biomarkers in head and neck squamous cell carcinoma cancer patients (HNSCC-CPs), but differences between HNSCC-CPs and healthy adults (HAs) are insufficiently described. Methods: 10-color flow cytometry (FCM) was used for in-depth immunophenotyping of PB samples of 963 HAs and 101 therapy-naïve HNSCC-CPs. Absolute (AbsCC) and relative cell counts (RelCC) of leukocyte subsets were determined. A training cohort (TC) of 43 HNSCC-CPs and 43 HAs, propensity score (PS)-matched according to age, sex, alcohol, and smoking, was used to develop a score consecutively approved in a validation cohort (VC). Results: Differences in AbsCC were detected in leukocyte subsets (*p* < 0.001), but had low power in discriminating HNSCC-CPs and HAs. Consequently, RelCC of nine leukocyte subsets in the TC were used to calculate 36 ratios; receiver operating characteristic (ROC) curves defined optimum cut-off values. Binary classified data were combined in a score based on four ratios: monocytes-to-granulocytes (MGR), classical monocytes-to-monocytes (clMMR), monocytes-to-lymphocytes (MLR), and monocytes-to-T-lymphocytes (MTLR); ≥3 points accurately discriminate HNSCC-CPs and HAs in the PS-matched TC (*p* = 2.97 × 10^−17^), the VC (*p* = 4.404 × 10^−178^), and both combined (*p* = 7.74 × 10^−199^). Conclusions: RelCC of leukocyte subsets in PB of HNSCC-CPs differ significantly from those of HAs. A score based on MGR, clMMR, MLR, and MTLR allows for accurate discrimination.

## 1. Introduction

Head and neck squamous cell carcinoma (HNSCC) is the sixth most common malignant tumor in the world [1,2]. The development of HNSCC is strongly associated with tobacco and high-level alcohol consumption as well as infection with high-risk subtypes of the human papillomavirus (HPV). Particular human leukocyte antigen (HLA) alleles and haplotypes have been linked to the development of HNSCC and prognosis independent of these risk factors [3,4]. Several studies investigated the absolute (AbsCC) and relative cell count (RelCC) and the change in leukocyte subsets that occurred in association with the disease. As these subsets are known to play a key role in immune surveillance and anti-tumor immune responses, which are dependent on the effective cooperation of monocyte-derived antigen-presenting cells (APC) in the stimulation of cytotoxic (Tc, CD3^+^CD8^+^) and helper T-cells (Th, CD3^+^CD4^+^), AbsCC and RelCC of these leukocytes are thought to be critical in the development of HNSCC. For example, a low level of invariant natural killer T cells (iNKT) is associated with a poor outcome [5]. Regulatory T-cells (Treg, CD3^+^CD4^+^CD25^+^) mediate immune suppression by secreting interleukin (IL-) 10 and transforming growth factor (TGF-) *β* to provide a helpful tolerogenic environment for tumor development and progression [6,7]. Absolute numbers of neutrophils, lymphocytes, and monocytes are higher in HNSCC and venous blood [6,7,8]; high numbers of neutrophils indicate a poor prognosis in several cancers, including HNSCC, because of their tumor-promoting activities, such as migration, angiogenesis, invasion, metastasis, immunosuppression, and mutagenesis [8,9,10]. A recent study showed the frequency of CD14^+^CD16^−^HLA-DR^high^ monocytes to be a strong predictor of progression-free (PFS) as well as overall survival (OS) in response to anti-PD-1 immunotherapy with the immune-checkpoint inhibitor; a clear response to the treatment was also shown in the T-cell compartment [11]. Ratios of various leukocyte subsets have been shown to be strong survival predictors in HNSCC. A high lymphocyte-to-monocyte ratio (LMR) or, vice versa, a low monocyte-to-lymphocyte (MLR) ratio in peripheral blood, were proven to be associated with a good prognosis in several HNSCC studies [12,13], whereas a high neutrophil-to-lymphocyte (NLR) ratio, as well as a high platelet-to-lymphocyte (PLR) ratio, corresponded with decreased disease-free survival (DFS) and OS [13,14].

Whereas the majority of these studies aimed at linking changes in leukocyte subsets to the stage of the disease (early vs. advanced, with or without loco-regional or distant metastasis), localization of the primary lesion as well as the T, N, and M categories, our aim was to detect how subsets of leukocytes in blood samples of HNSCC-cancer patients (HNSCC-CPs) differ from those taken from healthy adults (HAs). Moreover, we aimed to elucidate if differences may allow for discriminating HNSCC-CPs and HAs, potentially by building a score that summarizes these differences.

We hypothesized that high-throughput measurement utilizing 10-color flow cytometry (FCM) and parallel enumeration of leukocyte subsets in a large population-based HA cohort and a cohort of HNSCC-CPs would allow for reliable quantification of leukocyte subsets and calculation of MLR, NLR, and other ratios to discriminate HNSCC-CPs from HAs. Earlier studies suffer from potential contamination of leukocyte subsets caused by the earlier unavoidable technical limitations. These limitations resulted from low throughput and the use of two or three fluorescence-detection channels only. This led to small numbers of simultaneously analyzable staining patterns and, hence, a low number of enumerable leukocyte subsets in a single run. Unfortunately, this caused the error-prone need to combine multiple runs to do the immunophenotyping. Benefitting from advances in flow cytometry, especially high-throughput measurement utilizing 10-color FCM and parallel enumeration of leukocyte subsets by use of an optimized multicolor immunofluorescence panel (OMIP) consisting of 13 monoclonal antibodies to overcome the discussed limitations (OMIP-023 [15]), we were able to analyze a total of 15 leukocyte subsets simultaneously. Based on these high-quality data, we are able to show differences between HNSCC-CPs and HAs and present a leukocyte score that focuses on the most significant altered ratios in HNSCC-CPs able to discriminate both.

## 2. Results

The CONSORT diagram (Figure 1) shows information about selection of patients. 

Eligible for final analysis were 101 HNSCC-CPs as well as 963 HAs. First, the AbsCCs of the different leukocyte subsets of HNSCC-CPs and HAs were compared (Table 1).

The following leukocyte subsets were significantly different in numbers per milliliter: granulocytes (SSC^high^), neutrophils (SSC^high^CD16^+^), monocytes (SSC^med^CD14^high^), classical monocytes (SSC^med^CD14^high^CD16^dim^), total lymphocytes (SSC^low^), T-lymphocytes (CD3^+^CD16^−^CD56^−^), and T-cell subpopulations (CD3^+^CD8^+^ cytotoxic T-cells, CD3^+^CD4^+^ T-helper cells, and CD3^+^CD4^+^CD8^+^ double-positives).

### 2.1. Calculation of RelCC and PS-Matching

Since the AbsCC had low power regarding discrimination of HNSCC-CPs and HAs, the RelCCs of nine leukocyte subsets (granulocytes, neutrophils, lymphocytes, T-lymphocytes, T-helper cells, cytotoxic T-cells, monocytes, classical monocytes, and nonclassical monocytes) were calculated and differed significantly (Table 2).

As shown by Melzer et al. [16], sex and age affect AbsCCs and RelCCs of leukocyte subsets in venous blood. Risk factors linked to development and outcome of HNSCC, including, e.g., high-level intake of alcohol and smoking, might also interfere with AbsCC and RelCC. Of the HAs and HNSCC-CPs, 59.1% and 15.8% were nonsmokers; the mean of 5.94 and 32.16 pack years, respectively, also differed significantly (Table 2). Consequently, propensity score (PS)-matching was used to identify a training cohort (TC) of 43 HAs and 43 HNSCC-CPs without significant differences in risk factors (all *p* ≥ 0.4830). All other HAs and HNSCC-CPs were used as a validation cohort (VC). Besides the characteristics regarding the risk factors sex, age, alcohol, and tobacco smoking, Table 2 shows RelCCs in TC, VC, and both cohorts combined.

### 2.2. Receiver Operating Characteristics and Cut-Offs for Discriminating HAs and HNSCC-CPs

Whereas mostly classical monocytes were found in HAs, the intermediate monocytes dominated in HNSCC-CPs, and RelCCs of nonclassical monocytes were low in both (Figure 2a). Subsequently, the ratios of all RelCCs were permutated and revealed within the 36 possible ratios the ratios of monocytes-to-granulocytes (MGR), classical monocytes-to-monocytes (clMMR), monocytes-to-lymphocytes (MLR), and monocytes-to-T-lymphocytes (MTLR) as those achieving the highest level of significance (Figure 2b).

The percentage of classical (SSC^med^CD14^high^CD16^dim^) and nonclassical monocytes (SSC^med^CD14^dim^CD16^high^) demonstrated the highest significance and a nearly dichotomous distribution (Figure 2a).

Thereafter, receiver operating characteristic (ROC) curves were used to identify optimum cut-off values for prediction of being an HNSCC-CP. Figure 3 shows the four ratios that achieved the highest level of significance: MGR, clMMR, MLR, and MTLR according to the areas under the curve (AUC) and the corresponding level of significance. These ratios achieved an AUC of 95.6% (CI 95% 91.8–99.4%; *p* = 6.06∙× 10^−13^), >99.9% (CI 95% 100.0–100.0%; *p* = 3.07∙× 10^−15^), 75.7% (CI 95% 65.1–86.2%; *p* = 5.14∙× 10^−5^), 98.4% (CI 95% 96.3–100.0%; *p* = 2.37∙× 10^−14^) in the TC of 43 HNSCC-CPs and 43 HAs, and 97.4% (CI 95% 94.2–100.0%; *p* = 6.69∙× 10^−35^), 99.8% (CI 95% 99.7–100.0%; *p* = 2.51∙× 10^−38^), 98.3% (CI 95% 97.3–99.3%; *p* = 1.40∙× 10^−35^), and 97.3% (CI 95% 96.0–98.7%; *p* = 3.18∙× 10^−34^), respectively, in the validation cohort (VC) of 58 HNSCC-CPs and 920 HAs (Figure 3).

### 2.3. Development of a Leukocyte Score to Identify HNSCC-CPs

The significantly different RelCCs of therapy-naïve HNSCC-CPs as compared to PS-matched HAs allowed for the definition of a score to sum up the number of ratios above the ROC-curve-derived cut-offs. Based on increased RelCCs of monocytes in HNSCC-CPs, the cut-offs for optimum binary classification of HAs are MGR ≤ 0.1110, MLR ≤ 0.4040, MTLR ≤ 0.6283, and clMMR ≥ 0.6733. Classification of being an HNSCC-CP (that is, MGR, MLR, or MTLR above cut-off and clMMR below cut-off) scored 1. Subsequently, the values achieved for the four scores were summed up, yielding a maximum of four points whenever all ratios pointed to being an HNSCC-CP. A score of ≥3 points was determined as the optimum cut-off for identifying HNSCC-CPs, whereas scores <3 defined HAs. The score was then tested on a VC of 920 HAs and 58 HNSCC-CPs. After that, the score was applied to both cohorts combined.

Applying the leukocyte score as new classifier, a sufficient specificity of >88% and sensitivity of >90% were reached in all three cohorts (Figure 4). In the TC, all HAs and 39 of 43 of the HNSCC-CPs were classified correctly as well as 913 of 920 HAs and 54 of 58 HNSCC-CPs in the VC and 956 of 963 HAs and 93 of 100 HNSCC-CPs in the combined cohort. HNSCC-CPs and HAs could be discriminated in the PS-matched TC (*p* = 2.97 × 10^−17^), the VC (*p* = 4.04 × 10^−178^), and combined (*p* = 7.74 × 10^−199^). Since the number of participants in the HA cohort was much bigger than the number of HNSCC-CPs, the bias- and prevalence-independent measures, the score’s Informedness and Markedness were calculated [17] and revealed Informedness and Markedness of >90% and >88%, respectively, in all three cohorts (Figure 4). Bootstrapping utilizing the bias-corrected accelerated (BCa) method revealed significant values for Lambda (0.860, BCa 95% CI 0.780–0.928; *p* < 10^−18^), Goodman and Kruskal Tau (0.851, BCa 95% CI 0.772–0.923; *p* = 1.19 × 10^−198^), and the uncertainty coefficient (0.798, BCa 95% CI 0.711–0.885; *p* = 3.81 × 10^−117^). Cramer-V (0.923, BCa 95% CI 0.879–0.961; *p* = 7.74 × 10^−199^), Phi (0.923, BCa 95% CI 0.879–0.961; *p* = 7.74 × 10^−199^), and the contingency coefficient *C* (0.678, BCa 95% CI 0.660–0.693; *p* = 7.74 × 10^−199^) equally demonstrate a good performance of the score. Cross-validation within the Mantel–Haenszel statistics to estimate conditional probability revealed significance for the combined odds ratio (3254.9, BCa 95% CI 613.1–17,280.0; *p* = 2.19 × 10^−21^) even on the natural logarithm scale (ln OR = 8.088, BCa 95% CI 6.797–10.030; *p* = 1.06 × 10^−3^). The a-posteriori (Bayes) probability *P* (HNSCC-CP|leukocyte score ≥3) was found to be 100%, 88.5%, and 93.0% in TC, VC, and in both cohorts combined, respectively.

### 2.4. Classification Characteristics of the Leukocyte Score

Table 3 shows the characteristics of correctly and incorrectly classified HNSCC-CPs. Given the low number of only seven (7%) of the 100 blood samples that were not correctly classified as from HNSCC-CP, neither sex, age, localization, Tumor-Node-Metastasis (TNM) category, nor etiologically relevant risk factors, such as smoking history, alcohol consumption, or the HPV status in oropharynx cancer, demonstrated an association with reduced accuracy. Moreover, the leukocyte score demonstrates a potential to also classify blood samples from laryngeal HNSCC-CPs with small-volume disease, including four patients with pTisN0M0 and four with pT1aN0M0, correctly. However, two samples from pT1aN0M0 vocal cord carcinoma and a pT1N0M0 tongue carcinoma patient were among the seven misclassified samples. The only misclassified sample from a T4 cancer was from a male never-smoker of age 88 having a pT4N0M0 hypopharynx cancer, whereas the other misclassified samples were from a young HNSCC-CP without positive neck nodes. There was only one misclassified sample from a HNSCC-CP with multiple neck nodes (from a male heavy drinker with pT3pN2bcM0 tonsillar cancer).

## 3. Discussion

Within the framework of the LIFE cohort studies (information is provided in 4.1. Subjects), this prospectively defined nested case-control study demonstrates significant differences in AbsCC (Table 1) and RelCC (Table 2) of leukocyte subsets in peripheral blood between HAs and pre-treatment HNSCC-CPs using the validated OMIP-023 antibody cocktail and simultaneous analyses of 15 leukocyte subsets applying 10-color FCM [15,16]. Both the AbsCC and the RelCC show significant alterations, especially in the monocytes, classical monocytes, lymphocytes, T-lymphocytes, and granulocytes. Differences between HAs and HNSCC-CPs are mostly discussed as effects of higher age and the strongly increased exposure or history of exposure of HNSCC-CP to alcohol and tobacco smoke and the inflammation accompanying these risk factors for HNSCC. Therefore, a PS-matching considering history of tobacco smoking (in pack-years), alcohol consumption (in grams per day), male sex, and age was implemented. A caliper width of 0.2 standard deviations of the linear predictor was used to perform a 1:1 matching as this is an optimal matching algorithm [20] and recommended for PS-matching by Kuss et al. [21].

As the alterations in RelCC are observed despite PS-matching regarding known risk-factors (increased age, male sex, tobacco smoking, and high-level alcohol consumption; Table 2), the differences in AbsCC and RelCC are suggested to be linked rather to the disease than the named risk factors. By permutation of the RelCC of nine leukocyte subsets with significant differences, we were able to discover the four ratios within the 36 combinations demonstrating the highest discriminative power of HAs and HNSCC-CPs based on the individual ratios by using ROC analyses. These four ratios are MGR, MLR, MTLR, and clMMR (Figure 3). They can be combined in a leukocyte score.

The bias- and prevalence-independent measures Markedness [17] of >88% and Informedness [17] of >90% in TC, VC, and both cohorts combined demonstrate the score’s suitability to detect HNSCC-CPs (Figure 4). This is verified by the ability of the score to discriminate HNSCC-CPs from HAs with and without a high risk factor profile. The score detected HNSCC-CPs even in early stages (UICC 0, I, and II) very reliably (28 of the 31 HNSCC-CPs were classified correctly; Table 3). Given the low number of only seven misclassified HNSCC-CPs, we found only insignificant differences with respect to age, sex, localization of the primary lesion, TNM categories and stage, as well as smoking, alcohol consumption, and HPV-driven disease versus others (Table 3).

Suffering from HNSCC is reported to be associated with systemic inflammation marked by an increased number of monocytes and neutrophils in venous blood [22]. Neutrophils of HNSCC-CPs show reduced spontaneous apoptosis compared to HAs; the level of associated cytokines is significantly higher in venous blood of HNSCC-CPs compared to HAs [23]. A lower lymphocyte count and, thus, a higher MLR and NLR correlate with the clinical tumor stage in oropharyngeal squamous cell carcinoma (OPSCC) [24]. T-lymphocytes are suppressed in HNSCC due to, inter alia, an increase of CD14^+^ HLA-DR^−^ myeloid-derived suppressor cells (MDSCs) in peripheral circulation [25]. Lymphopenia in general is an indicator of poor outcome, as lymphocytes play an important role in the anti-tumor immune response [26]. Taken together, numerous studies demonstrated particular ratios, including MLR (or LMR), NLR, and other ratios, to be linked to an altered outcome and even as independent predictors for survival of HNSCC-CPs with primary HNSCC in the larynx, hypopharynx, oropharynx, and the oral cavity [8,27,28]. Unsurprisingly, the meta-analyses also found significant prognostic effects of the MLR and other ratios on outcome [28]. However, these ratios observed in HNSCC-CPs were never compared in a sufficient way with those in HAs; in particular, healthy persons with or without identical patterns of the “classical” risk factors for HNSCC. The framework of the LIFE study allowed us to overcome these limitations and demonstrates that the altered ratios are useful not only for prognostic classification of HNSCC-CPs but also as a biomarker for bias-free detection of HNSCC-CPs.

The findings in this study coincide with current studies, as they also show elevated monocytes, classical monocytes, granulocytes, and neutrophils as well as a decrease in lymphocytes and T-lymphocytes in venous blood in HNSCC-CPs in comparison with HAs. The ratios reflect the interaction between the different immune cells and the changes in the immune system over the immune-editing course of HNSCC from elimination and equilibrium to escape [8,27,28].

Our study has limitations. Based on a limited budget and the given time-frame, the HNSCC-CPs of the sub-study LIFE B7 head and neck cancer could not be analyzed completely and 10-color FCM using OMIP-023 was restricted to 161 consecutive patients. Of these, only 101 gave datasets from therapy-naïve primary HNSCC-CPs without synchronous or meta-synchronous diagnosed malignancy of other histology (Figure 1). The pattern of risk factors linked to development of HNSCC required a PS-matching of HAs and HNSCC-CPs to identify the four ratios differing significantly between both and being able to discriminate them in the TC. Combining these potential classifiers allowed for building a score. This part of our investigation is, therefore, to be categorized as a hypothesis-generating retrospective analysis. The validation of the score implied the VC that was not independent but rather included the subsample of HNSCC-CPs and HAs not included in the TC. Moreover, one of the 58 HNSCC-CPs was not included in the validation of the score, because the data about the lymphocytes and T-lymphocytes were missing. The two ratios that were known for this patient were higher than the cut-off values and, thus, showed a trend to be classified as an HNSCC-CP. However, no HA control had to be excluded, and, therefore, the comparison of AbsCC, RelCC, and the ratios as well the thereof-derived score is based on a sufficient number of cases. The extension of our analyses to bias- and prevalence-independent measures adds further evidence to the potential benefit of 10-color FCM regarding detection of HNSCC-CPs. Moreover, Bayesian statistics indicated the high a-posteriori probability of being an HNSCC-CP whenever the leukocyte score is ≥3: *P* (HNSCC-CP|leukocyte score ≥3) in TC, VC, and in both cohorts combined were 100%, 88.5%, and 93.0%, respectively. The few misclassified HNSCC-CP cases do not allow for a detailed analysis of misclassification characteristics. However, small-volume disease without local or systemic metastasis is correctly identified, and the risk factors demonstrated no significant effect on the classification characteristics. As our case-control study included only a population-based sample of healthy adults from Leipzig without history or development of HNSCC to date, and we do not have any information about the development of other cancers since providing the blood sample for the here-included analyses, we have to face the well-known limitations of case-control studies.

Unfortunately, no data about the intra-tumoral present leukocyte subsets and their distribution in the HNSCC were available. Further studies should investigate if the here-described differences in peripheral blood can also be observed in the tumor micro-environment and are linked to particular gene expression patterns [2,14,29,30]. Also, further research is needed to investigate possible connections between TNM categories, tumor size, and lymph node involvement and leukocyte cell count, MGR, MLR, MTLR, clMMR, and the score. A follow-up study is necessary to investigate the further possible change in leukocyte subsets throughout therapy and after curative treatment.

## 4. Materials and Methods

### 4.1. Subjects

Within the prospective population-based study LIFE, a total of 1064 blood samples were taken after obtaining written informed consent and underwent 10-color FCM analysis (see below). The samples were from two cohorts, 963 participants of LIFE Health Adult recruited at the LIFE study center (age 40 to 79 years, among them 352 with full risk factor anamnesis), and 101 consecutive HNSCC-CPs of the LIFE Head and Neck Cancer cohort [31] accrued between February 2011 and December 2011 (Figure 1). The latter blood samples, from 16 female and 85 male therapy-naïve HNSCC-CPs (age of 30 to 90 years) with pathohistologically confirmed HNSCC without history of malignancy of other histology, were obtained prior to treatment at the ENT Department of the University Hospital Leipzig. Details of the study design of LIFE have been published elsewhere [32]. There were no racial differences as all participants were of Caucasian ethnicity. All HNSCC-CPs and HAs provided written informed consent. Self-reported exposure to the risk factors alcohol and smoking was assessed using standardized questionnaires. The complete protocol of the LIFE study and both sub-studies was approved by the local Ethics Committee of the University Leipzig, Germany (votes 201-10-12072010 and 202-10-12072010).

### 4.2. Blood Sample Collection, Preparation, and Staining

Peripheral blood (PB) was collected in a 9-mL tube (Sarstedt, Germany) containing EDTA as an anticoagulant. Samples were collected in the morning from overnight-fasting participants and were processed within 2 h after drawing. Sample preparation was performed as described previously [15,16] Briefly, erythrocytes were lysed by adding to 1 mL of PB 25 mL of lysis buffer (8.3 g∙L^−1^ NH_4_Cl, 1 g∙L^−1^ KHCO_3_, and 0.04 g∙L^−1^ EDTA) for 10 min followed by washing twice with 15 mL of phosphate-buffered saline (PBS; Sigma–Aldrich, Deisenhofen, Germany) and resuspension in 200 µL of PBS supplemented with 5 g∙L^−1^ bovine serum albumin and 0.5 g∙L^−1^ sodium azide. Cells were then labeled with an Optimized Multicolor Immunofluorescence Panel (OMIP-023) [15] consisting of 13 antibodies labeled to 10 different fluorochromes [15,16]. Optimal antibody concentrations were determined by titration. The monoclonal antibodies were purchased from Beckman Coulter (BC: CA, USA) and Becton Dickinson Labware (BD: NJ, USA). The OMIP-023 consisted of CD3-KrO (T cells; BC B00068), CD4-APC-H7 (Th cells; BD 641398), CD8-FITC (Tc cells; BC A07756), CD14-FITC (LPS co-receptor on monocytes; BC IM0645U), CD45-PB (pan-leukocyte antigen; BC A74743), HLA-DR-APC (MHC-II; BC IM3635), CD69-PE (early-activation antigen; BC IM1943U), CD16-PE-Cy7 (Fc-γ receptor III on neutrophils, monocytes, NK; BC 6607118), CD19-FITC (B cells; BC A07768), CD25-ECD (IL2-Receptor, Treg; BC 6607112), CD56-PE-Cy7 (N-Cam on NK, NKT; BC A21692), CD38-PE-Cy5.5 (activated T and B cells; BC A70205), and CD127-APC-Ax700 (IL-7 receptor α chain, Treg; BC A71116). Aliquots of 100 µL leukocyte-enriched PB were stained with 2 µL of antibodies at room temperature for 2 h in the dark followed by fixation using 0.5% paraformaldehyde in PBS and starting the FCM within 4 h.

### 4.3. Flow Cytometry (FCM)

The total white blood cell count (WBC) was determined in all subjects using a Hematology Analyzer XN-9000 (Sysmex Europe GmbH, Norderstedt, Germany). FCM was performed on a Navios flow cytometer (Beckman Coulter, Pasadena, CA, USA), equipped with three lasers (405 nm, 40 mW; 488 nm, 22 mW; 638 nm; 25 mW) and 10 fluorescence detectors (blue laser: 525/40, 575/30, 620/30, 675/20, 695/30, and 755 LP; red laser: 660/20, 725/20, and 755 LP; and violet laser: 450/50, 550/40 nm). Color compensation values were obtained by single antibody stained cells using the automatic compensation control system of the Navios software. For each OMIP-023, analysis data of >500,000 events were collected. Flow cytometric data were analyzed using FlowJo V 7.6.4 (Tree Star, Asland, OR, USA) based on the OMIP-023 gating strategy [15].

### 4.4. Quality Control

The quality controls for the standardization of preanalytics, measurement, and data analysis were detailed elsewhere [15]. In short, for daily measurement, the Navios cytometer was calibrated with microbeads (Rainbow beads; Spherotech, Inc., Libertyville, IL, USA) and lasers were aligned with Flow Check Pro beads (BC). The premixed OMIP-023 antibody cocktail [15,16] was used within 5 days to ensure stability and to minimize pipetting errors. The reliability of manual gating was checked by analyzing identical samples by three evaluators with no significant bias and only low variance between the three readers (*R*^2^ = 0.993 to 0.999).

### 4.5. Manual Gating

Using the standardized gating protocol published in [15], gating was performed manually for each sample as follows (Appendix A): exclusion of air bubbles (time versus sideward scatter, SSC log: plot 1), non-single events (forward scatter, FCS time of flight versus FCS lin: plot 2) and CD45 events (anti-CD45 versus SSC log: plot 3). The CD45^+^ events were used to calculate percentages of leukocyte subpopulations. Absolute cell numbers were calculated from these percentages and the WBC count. As WBC counts of 28 HA were not available, AbsCC could be calculated for 935 (TC/VC: 43/892) only and 101 HNSCC-CPs (TC/VC: 43/58). According to SSC height, gates of granulocytes (Appendix A plot 1A: CD45^+^SSC^high^), monocytes (plot 1B: CD45^+^SSC^med^), and lymphocytes (plot 1C: CD45^+^SSC^low^) were discriminated and further subdivided. Neutrophils (plot 2: CD16^+^) and eosinophils (plot 2: CD16^−^) in granulocytes were discriminated. After excluding CD14^-^HLA-DR^-^ events (plot 3) and CD4^-^ events (plot 4) from monocyte analysis, classical (typical) monocytes (plot 5: CD14^++^CD16^+^) and nonclassical monocytes (plot 5: atypical (CD14^dim^CD16^++^) plus intermediate (CD14^+^CD16^++^)) were discriminated as well. Lymphocytes (plot 6) were gated into CD3^−^ (left; after exclusion of CD4^+^ events (plot 7) further analyzed in plot 8A: CD16/56^+^ NK cells; plot 8B: CD16/56^−^ B-lymphocytes) and CD3^+^ events (right; further analyzed in 9A: CD3^+^CD16/56^+^ NKT cells and 9B: CD3^+^CD16/56^−^ T-lymphocytes). Three T-lymphocyte subsets were differentiated (plot 10): CD8^high^ cytotoxic T cells (Tc), CD4^+^ T-helper cells (Th), and CD4^+^CD8^+^ double-positive T-cells (DPT). The gated T-helper cells were also used to identify CD25^+^ regulatory T-cells (Treg: anti-CD127 versus anti-CD25, plot 11). Exemplary data of OMIP-023-stained whole blood samples can be found on http://flowrepository.org/ with the Repository ID FR-FCM-ZZ74 and online (Appendix A).

### 4.6. Statistical Considerations and Propensity-Score Matching

As smoking, alcohol consumption, age, and male sex are prognostic factors for development of HNSCC and, therefore, were expected to be significantly higher in HNSCC-CPs, propensity-score matching (PS-matching) was pre-defined to be used to identify HAs within the total number of adults analyzed in the LIFE-A1 study. A 1:1 PS-matching was performed using SPSS version 24 (IBM, Amonk, NY, USA) with a caliper width of 0.2 standard deviations of the linear predictor [20,21].

### 4.7. Statistical Analyses

Numerical variables were compared between groups using heteroscedastic *t*-tests, whereas categorical variables in contingency tables were compared applying “classical” (biased) statistics to assess Pearson’s *χ*^2^, likelihood ratio, relative risk (RR) and 95% confidence interval (CI 95%), odds ratio (OR) and CI 95%, adjusted OR (OR#) and CI 95%, sensitivity, specificity, Youden index, Youden-score, inverse sensitivity, inverse specificity, Youden-score for inverse sensitivity and specificity, fall out, miss rate, accuracy, prevalence, and bias. We also calculated bias- and prevalence-independent measures BookmarkG [17], Bookmaker Informedness [17], Markedness [17], and Matthew’s correlation (the geometric mean of Bookmaker Informedness and Markedness) [17]. Cross-validation was executed using SPSS version 24 (IBM Corporation, Armonk, NY, USA) applying bootstrapping of 1000 iterations and the bias-corrected accelerated (BCa) method to validate the classification characteristics of the leukocyte score. *P* values below 0.05 were considered significant.

## 5. Conclusions

In-depth immunophenotyping of peripheral blood samples of HAs and therapy-naïve HNSCC-CPs using 10-color flow cytometry identifies the strongest differences in relative cell counts of monocytes, classical monocytes, lymphocytes, and T-lymphocytes. A score based on four leukocyte-subset ratios (clMMR, MLR, MTLR, and MGR) could be developed in a PS-matched TC of 86 (43 HNSCC-CPs vs. 43 HAs) and allows for highly sensitive and specific discrimination of HNSCC-CPs and HAs. The high predictive value of the score could be validated in a validation cohort of 58 HNSCC-CPs versus 920 HAs. This demonstrates the substantially different distribution of particular leukocyte subsets in peripheral blood of HNSCC-CPs. In conclusion, the four leukocyte-subset ratios (clMMR, MLR, MTLR, and MGR) are not only prognostic biomarkers as published earlier but represent biomarkers allowing for the discrimination of HNSCC-CPs and HAs.

## Figures and Tables

**Figure 1 cancers-11-00814-f001:**
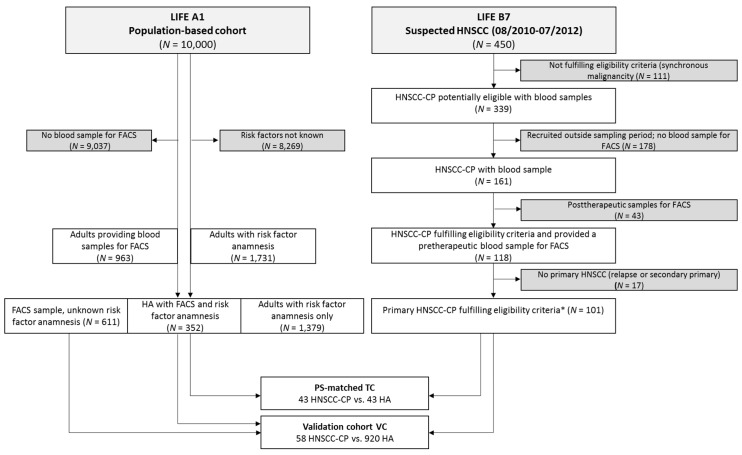
The CONSORT diagram. The selection of healthy adults (HAs, left) and head and neck squamous cell carcinoma cancer patients (HNSCC-CPs, right) are shown together with excluded samples by reason. Details regarding propensity score (PS)-matching to define the training cohort (TC) and the validation cohort (VC) can be found in the Methods section.

**Figure 2 cancers-11-00814-f002:**
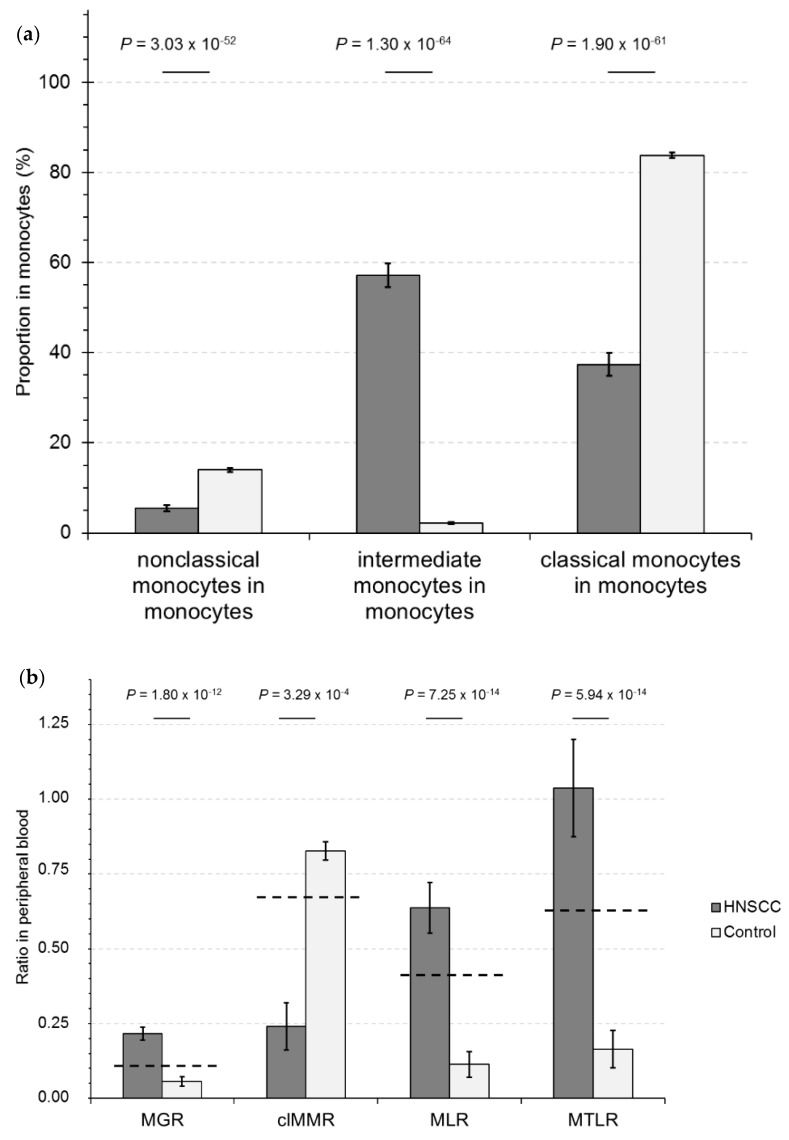
Relative cell counts (RelCCs) of monocyte subsets and ratios of RelCCs of leukocyte subsets defined by 10-color FCM using OMIP-023 [15,16] in peripheral blood (PB) of head and neck squamous cell carcinoma cancer patients (HNSCC-CPs) and healthy adults (control) in the propensity-score-matched (PS-matched) training cohort (TC) of 43 HNSCC-CPs and 43 healthy adults (control). (**a**) RelCCs shown for nonclassical monocytes in monocytes; intermediate monocytes in monocytes; classical monocytes in monocytes; (**b**) ratios of RelCCs of granulocytes to monocytes, monocytes to classical monocytes, lymphocytes to monocytes, and T-lymphocytes to monocytes. The corresponding levels of significance in heteroscedastic *t*-tests are shown.

**Figure 3 cancers-11-00814-f003:**
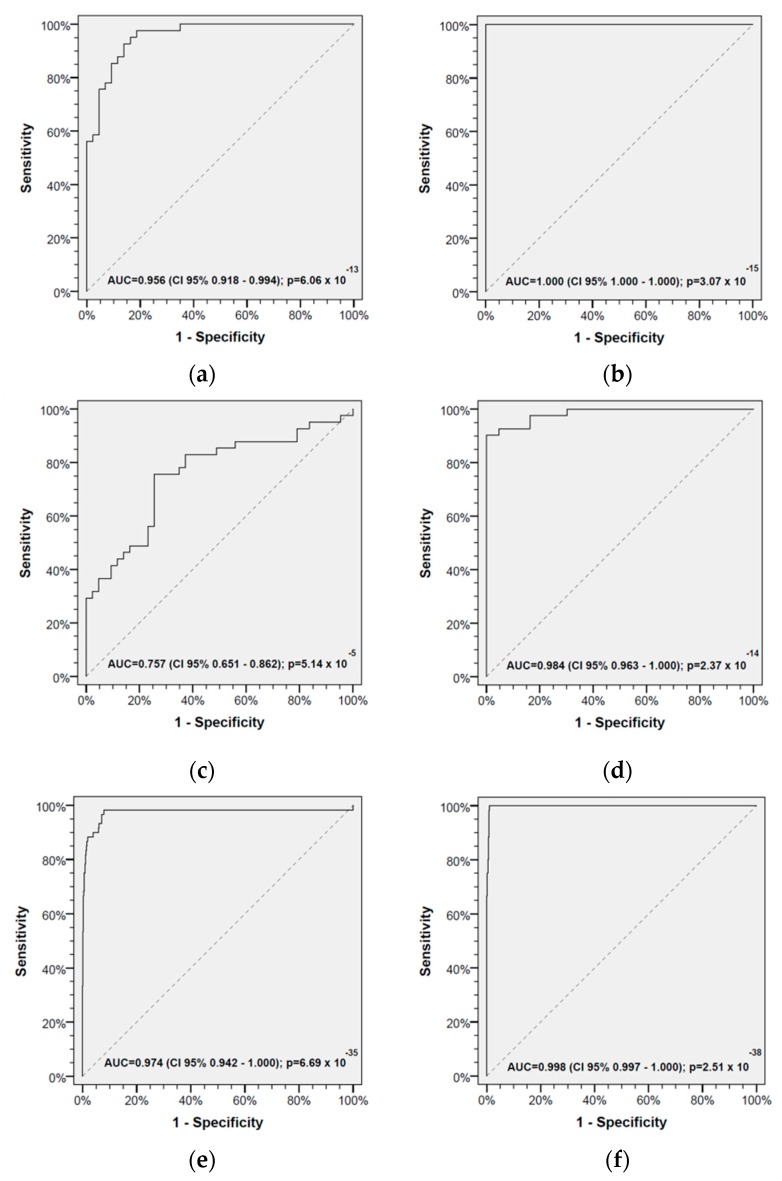
Receiver operating characteristic (ROC) curves for predicting being a head and neck squamous cell carcinoma cancer patient (HNSCC-CP) by the ratio of relative cell counts (RelCCs). (**a**,**e**) Ratio of granulocytes to monocytes; (**b**,**f**) ratio of classical monocytes to monocytes; (**c**,**g**) ratio of lymphocytes to monocytes; (**d**,**h**) ratio of T-lymphocytes to monocytes in (**a**–**d**) the propensity-score-matched (PS-matched) training cohort (TC) of 43 HNSCC-CPs and 43 healthy adults (HAs; left panel), and (**e**–**h**) the validation cohort (VC) of 58 HNSCC-CPs and 920 HAs (right panel). Areas under the curve (AUC), the 95% confidence intervals (CI 95%), and the corresponding level of significance are shown.

**Figure 4 cancers-11-00814-f004:**
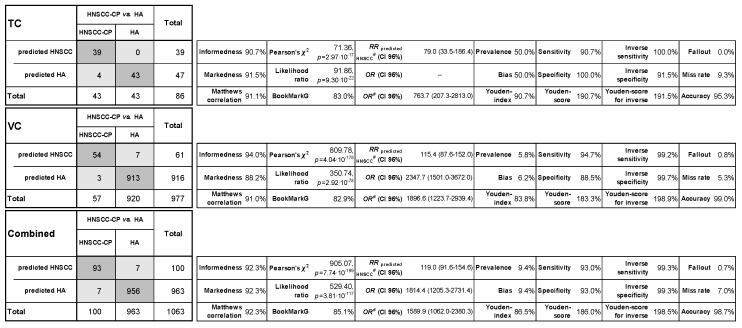
Binary contingency tables for the training cohort (TC) of 86 propensity-score-matched head and neck squamous cell carcinoma cancer patients (HNSCC-CPs) and healthy adults (HAs), the validation cohort (VC) of 977 cases (57 HNSCC-CPs and 920 HAs), and the combined cohorts (Combined) among 1063 blood samples classified by the leukocyte score. Shade coding indicates correct (gray) vs. incorrect (light gray) decisions with the thereof-derived Bookmaker Informedness [17], Markedness [17], Matthew’s correlation (the geometric mean of Bookmaker Informedness and Markedness) [17], relative risk (RR) and 95% confidence interval (CI 95%), odds ratio (OR) and CI 95%, adjusted OR (OR#) and CI 95%, Pearson’s χ^2^, likelihood ratio, BookmarkG [17] along with prevalence, bias, Youden index, sensitivity, specificity, Youden-score, inverse sensitivity, inverse specificity, Youden-score for inverse sensitivity and specificity, fall out, miss rate, and accuracy. # Adjusted according to Cox [18] and Moses et al. [19] by adding 0.5 to each cell to reduce bias and prevent undefined division by zero caused by empty cells.

**Table 1 cancers-11-00814-t001:** The mean and 95% confidence intervals (CI) for absolute cell counts (AbsCC) in 10^9^∙L^−1^ of leukocyte subsets in peripheral blood detected by 10-color flow cytometry utilizing the optimized multicolor immunofluorescence panel (OMIP)-023 [15] in healthy adults (HAs) and HNSCC cancer patients (HNSCC-CPs) in the propensity-score-matched training cohort (TC), the validation cohort (VC), and both combined. Significant differences are shown in bold.

Leukocyte Subsets	Marker	Healthy Adults (HAs) Mean (95% CI)		HNSCC Patients (HNSCC-CPs) (95% CI)		*p*-Value *
**Combined ^a^**	***n***	**935**		**101**		
Granulocytes	SSC^high^	3.93	(3.84–4.02)	4.81	(4.41–5.22)	**0.0001**
Neutrophils	SSC^high^CD16^+^	3.64	(3.55–3.73)	4.67	(4.27–5.07)	**<0.0001**
Eosinophils	SSC^high^CD16^−^	0.14	(0.13–0.15)	0.14	(0.12–0.16)	0.8979
Monocytes	SSC^med^CD14^high^	0.26	(0.25–0.27)	0.36	(0.33–0.40)	**<0.0001**
nonclassical m.	SSC^med^CD14^dim^CD16^high^	0.03	(0.03–0.04)	0.04	(0.03–0.04)	0.3414
classical m.	SSC^med^CD14^high^CD16^dim^	0.22	(0.21–0.23)	0.32	(0.29–0.35)	**<0.0001**
Lymphocytes	SSC^low^	1.77	(1.73–1.81)	1.01	(0.91–1.10)	**<0.0001**
B-cells	CD3^−^CD16/CD56^−^ CD19^+^	0.19	(0.18–0.20)	0.17	(0.15–0.20)	0.2814
T-cells	CD3^+^CD16/CD56^−^	1.23	(1.20–1.26)	0.91	(0.82–0.99)	**<0.0001**
Tc cells	CD3^+^CD8^+^	0.24	(0.23–0.26)	0.19	(0.16–0.22)	**0.0011**
DPT	CD3^+^CD4^+^CD8^+^	0.02	(0.02–0.02)	0.01	(0.01–0.02)	**<0.0001**
Th cells	CD3^+^CD4^+^	0.89	(0.86–0.92)	0.75	(0.67–0.82)	**0.0005**
regulatory T-cells ^e^	CD3^+^CD4^+^CD25^+^CD127^+^	0.07	(0.07–0.07)	0.07	(0.06–0.08)	0.5573
NKT cells	CD3^+^CD8^+^ CD16/CD56^+^	0.10	(0.10–0.11)	0.10	(0.08–0.12)	0.8046
NK cells	CD3^−^CD8^−^ CD16/CD56^+^	0.31	(0.30–0.32)	0.29	(0.25–0.32)	0.2452
**PS-matched TC ^b^**	***n***	**43**		**43**		
Granulocytes	SSC^high^	3.84	(3.41–4.28)	4.69	(4.05–5.32)	**0.0370**
Neutrophils	SSC^high^CD16^+^	3.66	(3.24–4.08)	4.56	(3.94–5.19)	**0.0234**
Eosinophils	SSC^high^CD16^−^	0.17	(0.12–0.21)	0.12	(0.09–0.16)	0.1334
Monocytes	SSC^med^CD14^high^	0.31	(0.28–0.34)	0.37	(0.31–0.42)	0.0838
nonclassical m.	SSC^med^CD14^dim^CD16^high^	0.03	(0.03–0.04)	0.04	(0.03–0.05)	0.3462
classical m.	SSC^med^CD14^high^CD16^dim^	0.27	(0.24–0.30)	0.32	(0.27–0.37)	0.1235
Lymphocytes	SSC^low^	1.84	(1.66–2.02)	1.02	(0.89–1.15)	**<0.0001**
B-cells	CD3^−^CD16/CD56^−^ CD19^+^	0.21	(0.17–0.24)	0.17	(0.14–0.20)	0.0979
T-cells	CD3^+^CD16/CD56^−^	1.20	(1.08–1.33)	0.91	(0.79–1.03)	**0.0015**
Tc cells	CD3^+^CD8^+^	0.20	(0.17–0.24)	0.20	(0.15–0.25)	0.9290
DPT	CD3^+^CD4^+^CD8^+^	0.01	(0.01–0.02)	0.02	(0.01–0.02)	0.4755
Th cells	CD3^+^CD4^+^	0.92	(0.81–1.02)	0.73	(0.63–0.83)	**0.0147**
regulatory T-cells	CD3^+^CD4^+^CD25^+^CD127^+^	0.08	(0.07–0.09)	0.07	(0.06–0.08)	0.2114
NKT cells	CD3^+^CD8^+^ CD16/CD56^+^	0.09	(0.07–0.10)	0.11	(0.07–0.14)	0.2775
NK cells	CD3^−^CD8^−^ CD16/CD56^+^	0.40	(0.32–0.47)	0.29	(0.23–0.34)	**0.0222**
**VC ^c^**	***n***	**892**		**58**		
Granulocytes	SSC^high^	3.93	(3.84–4.03)	4.91	(4.39–5.43)	**0.0007**
Neutrophils	SSC^high^CD16^+^	3.64	(3.55–3.73)	4.75	(4.23–5.27)	**0.0001**
Eosinophils	SSC^high^CD16^−^	0.14	(0.13–0.15)	0.15	(0.13–0.18)	0.2848
Monocytes	SSC^med^CD14^high^	0.26	(0.25–0.26)	0.36	(0.32–0.40)	**<0.0001**
nonclassical m.	SSC^med^CD14^dim^CD16^high^	0.03	(0.03–0.04)	0.04	(0.03–0.04)	0.5644
classical m.	SSC^med^CD14^high^CD16^dim^	0.22	(0.21–0.22)	0.32	(0.28–0.36)	**<0.0001**
Lymphocytes	SSC^low^	1.76	(1.72–1.81)	1.00	(0.87–1.13)	**<0.0001**
B-cells	CD3^−^CD16/CD56^−^ CD19^+^	0.19	(0.17–0.20)	0.18	(0.15–0.21)	0.4031
T-cells	CD3^+^CD16/CD56^−^	1.23	(1.20–1.27)	0.91	(0.79–1.03)	**<0.0001**
Tc cells	CD3^+^CD8^+^	0.25	(0.23–0.26)	0.18	(0.14–0.21)	**0.0011**
DPT	CD3^+^CD4^+^CD8^+^	0.02	(0.02–0.02)	0.01	(0.01–0.01)	**<0.0001**
Th cells	CD3^+^CD4^+^	0.89	(0.86–0.91)	0.76	(0.65–0.86)	**0.0221**
regulatory T-cells ^e^	CD3^+^CD4^+^CD25^+^CD127^+^	0.07	(0.07–0.07)	0.07	(0.06–0.08)	0.7916
NKT cells	CD3^+^CD8^+^ CD16/CD56^+^	0.10	(0.10–0.11)	0.09	(0.07–0.11)	0.3587
NK cells	CD3^−^CD8^−^ CD16/CD56^+^	0.30	(0.29–0.31)	0.29	(0.24–0.33)	0.4768

* *p* values from heteroscedastic *t* tests; ^a^ TC + VC; ^b^ Propensity-score-matched training cohort (TC); ^c^ Validation cohort (VC); ^d^ pack years of smoking history are calculated by multiplying years of self-reported tobacco smoking by the mean of reported cigarettes per day divided by 20; ^e^ one HA without information on Treg counts.

**Table 2 cancers-11-00814-t002:** The mean and 95% confidence intervals (CI) for relative cell counts (RelCC) of various leukocyte subsets in peripheral blood based on absolute cell counts (AbsCC) in 10^9^∙L^−1^ detected by 10-color flow cytometry utilizing the OMIP-023 panel [15] in healthy adults (HAs) and HNSCC cancer patients (HNSCC-CPs) in the propensity-score-matched training cohort (TC), the validation cohort (VC), and both combined. Significant differences are shown in bold.

Characteristics and Leukocyte Subsets	Unit	Healthy Adults (HAs) Mean (95% CI)		HNSCC Patients (HNSCC-CPs) Mean (95% CI)		*p*-Value *
**Combined ^a^**	***n***	**963**		**101**		
Male sex	%	42.3		84.2		**<0.0001**
Age	years	54.09	(52.70–55.48)	60.54	(58.54–62.53)	**<0.0001**
Tobacco smoking	pack years ^d^	5.94	(4.55–7.32)	32.16	(27.25–37.07)	**<0.0001**
Alcohol (g/d) 0	%	14.2		16.0		**<0.0001**
>0–30	%	72.4		27.0		
>30–60	%	10.2		27.0		
>60	%	3.1		30.0		
Cell subsets RelCC						
Granulocytes	%	64.85	(64.16–65.53)	69.24	(67.17–71.31)	**0.0001**
Neutrophils	%	60.04	(59.33–60.75)	63.42	(61.26–65.57)	**0.0044**
Lymphocytes ^e^	%	30.35	(29.67–31.04)	24.11	(22.02–26.19)	**<0.0001**
T-lymphocytes ^e^	%	21.04	(20.49–21.60)	14.91	(13.32–16.50)	**<0.0001**
Tc cells ^e^	%	4.15	(3.96–4.34)	2.31	(1.92–2.70)	**<0.0001**
Th cells ^e^	%	15.23	(14.81–15.65)	11.84	(10.54–13.14)	**<0.0001**
Monocytes ^f^	%	4.43	(4.32–4.53)	16.43	(15.46–17.40)	**<0.0001**
nonclassical m. ^f^	%	0.59	(0.56–0.61)	0.88	(0.76–1.01)	**<0.0001**
intermediate m. _f_	%	0.09	(0.08–0.10)	9.47	(8.60–10.33)	**<0.0001**
classical m. ^f^	%	3.74	(3.65–3.84)	5.89	(5.43–6.35)	**<0.0001**
**PS-matched TC ^b^**	***n***	**43**		**43**		
Male sex	%	100		100		>0.9999
Age	years	60.25	(57.15–63.36)	61.78	(58.86–64.70)	0.4830
Tobacco smoking	pack years^d^	23.36	(15.42–31.30)	26.05	(19.60–32.49)	0.6075
Alcohol (g/d) 0	%	16.3		18.6		0.9722
>0–30	%	37.2		32.6		
>30–60	%	34.9		37.2		
>60	%	11.6		11.6		
Cell subsets RelCC						
Granulocytes	%	62.82	(59.45–66.19)	68.46	(65.16–71.76)	**0.0215**
Neutrophils	%	59.86	(56.35–63.37)	63.03	(59.76–66.31)	0.1983
Lymphocytes	%	31.89	(28.73–35.06)	24.89	(21.48–28.31)	**0.0042**
T-lymphocytes	%	20.67	(18.62–22.72)	15.27	(12.79–17.74)	**0.0015**
Tc cells	%	3.50	(2.97–4.04)	2.53	(1.79–3.26)	**0.0381**
Th cells	%	15.68	(14.05–17.30)	11.96	(9.98–13.93)	**0.0056**
Monocytes ^f^	%	5.35	(4.73–5.97)	16.57	(14.88–18.27)	**<0.0001**
nonclassical m. ^f^	%	0.57	(0.46–0.68)	0.79	(0.62–0.96)	**0.0344**
intermediate m. ^f^	%	0.04	(0.00–0.08)	9.32	(7.83–10.81)	**<0.0001**
classical m. ^f^	%	4.74	(4.18–5.30)	6.01	(5.36–6.66)	**0.0048**
**VC ^c^**	***n***	**920**		**58**		
Male sex	%	34.3		72.4		**<0.0001**
Age	years	53.23	(51.73–54.73)	59.61	(56.90–62.32)	**0.0001**
Tobacco smoking	pack years ^d^	3.51	(2.69–4.34)	36.70	(29.79–43.61)	**<0.0001**
Alcohol (g/d) 0	%	13.9		14.0		**<0.0001**
>0–30	%	77.3		22.8		
>30–60	%	6.8		19.3		
>60	%	1.9		43.9		
Cell subsets RelCC						
Granulocytes	%	64.94	(64.24–65.64)	69.81	(67.14–72.48)	**0.0010**
Neutrophils	%	60.04	(59.32–60.77)	63.70	(60.81–66.59)	**0.0199**
Lymphocytes ^e^	%	30.28	(29.58–30.98)	23.51	(20.90–26.13)	**<0.0001**
T-lymphocytes ^e^	%	21.06	(20.49–21.63)	14.64	(12.56–16.72)	**<0.0001**
Tc cells ^e^	%	4.18	(3.99–4.37)	2.14	(1.73–2.56)	**<0.0001**
Th cells ^e^	%	15.21	(14.77–15.64)	11.75	(10.02–13.48)	**0.0004**
Monocytes	%	4.38	(4.27–4.49)	16.33	(15.17–17.48)	**<0.0001**
nonclassical m.	%	0.59	(0.56–0.61)	0.95	(0.77–1.13)	**0.0003**
intermediate m.	%	0.10	(0.09–0.11)	9.58	(8.55–10.6)	**<0.0001**
classical m.	%	3.70	(3.60–3.80)	5.80	(5.16–6.44)	**<0.0001**

* *p* values from heteroscedastic *t* tests; # *p* values from Pearson’s Chi-square (*χ*^2^) tests; ^a^ TC + VC; ^b^ Propensity-score-matched trainings cohort (TC); ^c^ Validation cohort (VC); ^d^ pack years of smoking history are calculated by multiplying years of self-reported tobacco smoking by the mean of reported cigarettes per day divided by 20; ^e^ one HNSCC-CP without information on lymphocyte counts; ^f^ two HNSCC-CPs without information on monocyte count.

**Table 3 cancers-11-00814-t003:** Classification characteristics of the leukocyte score according to various epidemiological and clinical characteristics of HNSCC cancer patients in the combined cohort. No significant differences were detected.

Characteristic	Category	Score ≥ 3		Score < 3		*p* Value *
		n	(%)	n	(%)	
Sex	Female	14	(15.1)	1	(14.3)	0.9562
	Male	79	(84.9)	6	(85.7)	
Age (years)	<50	12	(12.9)	2	(28.6)	0.3932
	50–59	37	(39.8)	3	(42.9)	
	60–69	23	(24.7)	--	--	
	>70	21	(22.6)	2	(28.6)	
Tobacco smoking (pack years)	never smoker	14	(15.1)	2	(28.6)	0.5024
	<30	29	(31.2)	1	(14.3)	
	>30	50	(53.8)	4	(57.1)	
Alcohol (g/d)	0	14	(15.1)	1	(14.3)	0.8724
	1–30	24	(25.8)	3	(42.9)	
	31–60	26	(28.0)	1	(14.3)	
	>60	28	(30.1)	2	(28.6)	
	Unknown	1	(1.1)	--	--	
HPV	p16+ HPV16+	16	(17.2)	--	--	0.2312
	p16− HPV−	77	(82.8)	7	(100.0)	
Localization	Oropharynx	31	(33.3)	3	(42.9)	0.8241
	Hypopharynx	10	(10.8)	1	(14.3)	
	Larynx	23	(24.7)	2	(28.6)	
	Other	29	(31.2)	1	(14.3)	
UICC stage 7th ed.	0	4	(4.3)	--	--	0.3545
	I	10	(10.8)	2	(28.6)	
	II	14	(15.1)	1	(14.3)	
	III	7	(7.5)	2	(28.6)	
	IVA	49	(52.7)	2	(28.6)	
	IVB	8	(8.6)	--	--	
	IVC	1	(1.1)	--	--	
Early vs. advanced	Advanced	65	(69.9)	4	(57.1)	0.5612
	Early	24	(25.8)	3	(42.9)	
	SINIII	4	(4.3)	--	--	
T category	Tis	4	(4.3)	--	--	0.8318
	1	18	(19.4)	3	(42.9)	
	2	27	(29.0)	2	(28.6)	
	3	14	(15.1)	1	(14.3)	
	4a	24	(25.8)	1	(14.3)	
	4b	3	(3.2)	--	--	
	X	3	(3.2)	--	--	
T; locally early vs. advanced	T0–T2	52	(55.9)	5	(71.4)	0.4240
	T3–T4	41	(44.1)	2	(28.6)	
N category	0	37	(39.8)	4	(57.1)	0.2369
	1	6	(6.5)	2	(28.6)	
	2a	1	(1.1)	--	--	
	2b	23	(24.7)	1	(14.3)	
	2c	22	(23.7)	--	--	
	3	4	(4.3)	--	--	
Neck nodes	N0	37	(39.8)	4	(57.1)	0.3679
	N+	56	(60.2)	3	(42.9)	

* *p* values from Pearson’s Chi-square (*χ*^2^) tests.

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
