# Peer review of "Discrimination of Head and Neck Squamous Cell Carcinoma Patients and Healthy Adults by 10-Color Flow Cytometry: Development of a Score Based on Leukocyte Subsets"

_cancers, 2019, doi:10.3390/cancers11060814_

Round 1

Reviewer 1 Report

The authors add data to existing knowledge that leukocytes ratios or absolute values differ between healty controls and head and neck cancer patients. However, this is not head and neck cancer specific. Please provide values according to Tumor stages in local, nodal +, and Systemic settings. Please show data for HPV + and neg. tumors.

 It would be interesting to see how T1N0M0 perform. I would not expect any effect on global (non specific) leukocytes subsets. 

Author Response

GW: Addressing both suggestions, we included the new Table 3 containing information regarding the correct classification of blood samples from HNSCC-CP including HPV-driven OPSCC and small volume pT1aN0M0 HNSCC. The score demonstrates applicability of the newly developed score for detection of HNSCC-CP in general (in majority independent from the stage of disease in our in regarding variety in stages and case numbers per stage small cohort). Contrary to our expectations, even blood samples from HNSCC-CP with small volume cancer are in majority identified correctly (please compare Table 3 and the new paragraph “2.4 Classification characteristics of the Leukocyte Score”).

So far we do not know if the changes in leukocyte subsets in the peripheral blood are a pre-disposing factor for HNSCC development or changes in relative leukocyte counts appear only associated with progression of the disease. Regarding the latter, many studies demonstrated differences between leukocyte ratios linked to progression of disease based on differences observed in the comparison of pre-therapeutic samples from HNSCC-CP of different stage. However, we do not know anything about the time-point at which the four ratios exceed their individual thresholds identified in our investigation, and if other (adapted) thresholds will demonstrate a much poorer discrimination (because only blood samples of higher stage will exceed these limits) as we aimed on distinguishing healthy adults from HNSCC-CP, not HNSCC-CP of different stages. For the latter purpose, to gain information about prognosis etc., other leukocyte subset ratios appear to be much better suitable (Tregs/Th1, iNKT/CD56+, …). Maybe this will be the topic of following papers…

As the new Table 3 and the new paragraph “2.4 Classification characteristics of the Leukocyte Score” represent an essential contribution to finally end up with a higher quality paper, we have to say THANK YOU VERY MUCH to reviewer #1 for his advice.

Reviewer 2 Report

The study appears very interesting and methodologically well developed. The results reported could open new scenarios in head and neck cancer diagnosis and  prognosis, if confirmed by other research groups.

However, depending on the site, head and neck tumors have a wide heterogeneity in the mechanisms underlying their onset and progression and consequently in prognosis.

It would therefore be useful for the authors to report the results in relation to the site of origin.

Author Response

Comments and Suggestions for Authors

The study appears very interesting and methodologically well developed. The results reported could open new scenarios in head and neck cancer diagnosis and  prognosis, if confirmed by other research groups.

GW: Many thanks for the positive review of our paper and the implications of our findings.

However, depending on the site, head and neck tumors have a wide heterogeneity in the mechanisms underlying their onset and progression and consequently in prognosis.

GW: Many thanks for the good comment regarding heterogeneity in etiology and outcome of HNSCC related to localization of the primary lesion.

It would therefore be useful for the authors to report the results in relation to the site of origin.

GW: Addressing this suggestion, we included also some information about the classification characteristic respective to localization of the primary lesion into the new Table 3 that holds information about the correct and incorrectly classified blood samples from HNSCC-CP (please compare Table 3 and the new paragraph “2.4 Classification characteristics of the Leukocyte Score” and our response to comments from the other reviewers).

Many thanks for the helpful comments!

Reviewer 3 Report

The paper is largely built on the foundation of the earlier and ongoing work of the authors. The authors report the scores formed based on leukocyte subsets and which discriminate healthy individuals from those suffering from head and neck carcinomas. What about other cancer types??? This should be at least discussed more thoroughly as the reader is now left with a conclusion that this is something specific for head and neck carcinomas.

Specific comments:

There is a strange sentence at the beginning of discussion, please remove.

Discussion contains a paragraph that is like results (starts row 231) and the result figures are placed within the discussion. Please, reorganize.

The language needs checking: for example neutrophils and eosinophils are incorrectly spelled in some occasions.

and the sentence Neutrophils of HNSCC-CP show reduced spontaneous apoptosis than HA is not correct. There are few others

Author Response

GW: Many thanks for careful reading. You are completely right, and we erased the sentence.

Discussion contains a paragraph that is like results (starts row 231) and the result figures are placed within the discussion. Please, reorganize.

GW: Many thanks for the recommendation. We rearranged the Figures and inserted the new Table 3 according to the advice given by all reviewers including reviewer #3.

The language needs checking: for example neutrophils and eosinophils are incorrectly spelled in some occasions.

GW: Many thanks for careful reading. We corrected the mistakes in the tables.

and the sentence Neutrophils of HNSCC-CP show reduced spontaneous apoptosis than HA is not correct. There are few others

GW: We corrected the respective sentence and checked the manuscript for clarity and errors hindering readability.

Following your helpful comments supported us very well to achieve a now much better paper. Many thanks!

Reviewer 4 Report

The manuscript by Weichmann et al. is a great example of how progress in technology can contribute to new strategies in clinical diagnostics. Using a previously developed 10-color flow cytometry panel the authors successfully generate a new biomarker for discrimination between patients with head and neck cancers and healthy individuals. This biomarker is based on immune subset ratios that were found to be significantly different in samples from therapy-naïve cancer patients as compared to those from healthy individuals.

The reviewer has however identified a few errors regarding the following issues that have to be revised:

1)    The panel named OMIP-023 appears firstly in Table 1, without any explanation or reference. The description of this abbreviation is made in Materials and methods, at the end of the manuscript. Readers of Cancers journal might not be familiar with this nomenclature, which deserves to be defined in the Results section or even better the reference 17 (Cytometry A, 2014) cited in the Introduction, because this is the report describing for the first time this panel and the entire manuscript describes a clinical application of its development.

2)    Table 1 descriptors state “mean (95% CI)” without a precise definition of what is measured. It would be advisable to add “no per mL” above this for absolute cell counts in table 1 and “relative no per mL” for relative cell count in table2.

3)    When enumerating the risk factors linked to HNSCC in section 2.1 of Results the authors fail to mention HPV infection, which emerged as a significant cause for oral cancer development in the past decade. It would be interesting to add a discussion about why the HPV-infection was not considered for PS-matching between healthy and HNSCC bearing individuals and whether this factor might affect significantly the immune cell counts. Interestingly, there is a reference 15 on this matter in the list that is not commented however in text.

4)    The first phrase in the Discussion should be erased (it is from the authors guidelines maybe…)

5)    References 14 and 15 are erroneously cited instead of 17&18 in the first paragraph of discussions. 

6)    A discussion about how these technical developments might be applied to other cancers would be very useful. How could this new biomarker proposed for “prognostic classification” and “bias-free detection of HNSCC cancer patients” differentiate between cancers? 

7)    An error appears in the Manual Gating section in Materials and Methods. Plot 1 (defined as Time vs SSC) and plot 2 (FSC TOF vs FSC Peak Lin) are missing from Supplementary Figure 1. They are also different from Plot 1 and 2 in Ref 17. This has to be corrected.

8)    Please correct the Reference list. It appears that the order in the list does not respect the order in the text and some references are not cited.

Author Response

1)    The panel named OMIP-023 appears firstly in Table 1, without any explanation or reference. The description of this abbreviation is made in Materials and methods, at the end of the manuscript. Readers of Cancers journal might not be familiar with this nomenclature, which deserves to be defined in the Results section or even better the reference 17 (Cytometry A, 2014) cited in the Introduction, because this is the report describing for the first time this panel and the entire manuscript describes a clinical application of its development.

GW: Many thanks for this very welcome suggestion. We cited the reference [15] (new enumeration of Bocsi et al.) into all appropriate legends and inserted the red highlighted phrase into the next-to-last sentence improving the understandability of the Introduction:

Benefitting from advances in flow-cytometry, especially high-throughput measurement utilizing 10-color FCM and parallel enumeration of leukocyte subsets by use of an optimized multicolor immunofluorescence panel (OMIP) consisting of 13 monoclonal antibodies overcoming the discussed limitations (OMIP-023 [15]), we were able to analyze a total of 15 leukocyte subsets simultaneously.

2)    Table 1 descriptors state “mean (95% CI)” without a precise definition of what is measured. It would be advisable to add “no per mL” above this for absolute cell counts in table 1 and “relative no per mL” for relative cell count in table2.

GW: Many thanks for this advice. We inserted the relevant information into the legend of Table 1 (“in 109L-1). We did not follow the suggestion for Table 2 in the suggested way but rather added “ based on absolute cell counts (AbsCC)  in 109L-1

3)    When enumerating the risk factors linked to HNSCC in section 2.1 of Results the authors fail to mention HPV infection, which emerged as a significant cause for oral cancer development in the past decade. It would be interesting to add a discussion about why the HPV-infection was not considered for PS-matching between healthy and HNSCC bearing individuals and whether this factor might affect significantly the immune cell counts.

GW: Many thanks for the comment. As we compared HNSCC-CP with healthy adults (HA), they have to the best of our knowledge no cancer including HPV-infected/-driven cancer(s) like cancer e.g. of the cervix uteri, anal or penis cancer. Ongoing HPV infection without the link to malignancy could not be assessed in the cohort of HA providing blood samples for 10-color FCM. We do not have information about HPV-DNA in circulating blood or other relevant information. Therefore the use of HPV as matching factor for HNSCC-CP and HA cannot be used. A matching of HNSCC-CP with and without HPV-driven disease was not done as we detected only 16 cases with p16+ HPV16+ (DNA+RNA+) oropharynx carcinomas and all were correctly classified by the leukocyte score.

Interestingly, there is a reference 15 on this matter in the list that is not commented however in text.

GW: Many thanks for this comment and the suggestion to include the HPV status into the analyses. We cited our earlier paper as those patients included in the present analyses were also under investigation in that paper (now ref. 31) and the respective votes from our ethics committee.

We did the analyses of a potential impact of HPV on classification characteristics but decided not to stress this very interesting question intensively as the results of the respective analyses for HPV-driven oropharynx cancer patients was not different from the other cases. Here are some details:

Given the random distribution in our prospectively accrued cohort we had only 16 HPV-driven oropharynx cancer patients (p16+ HPV16+ OPSCC) among the 100 HNSCC-CP under investigation. The huge uncertainty in statistics dealing with such a low number was from our point of view much too high and the reliability too low to come to conclusive results. As, moreover, these 16 cases were all correctly classified by the leukocyte score as HNSCC-CP, we found no (potentially expected) significant difference requiring an explanation. We found not the expected difference in leukocyte subsets between HPV-driven OPSCC and other HNSCC. This may be, however, only a result of the in this regard underpowered investigation comparing 16 vs. 84 cases and the huge amount of potentially relevant covariates. Therefore we rather preferred to skip potentially misleading statements about any HPV-associated effects on the distribution of leukocyte subsets and a (based on the case numbers too soon) speculation about a potential effect on the classification of HNSCC-CP by the leukocyte score. We had too few data.

As we now included the new Table 3 containing information regarding the correct classification of blood samples from HNSCC-CP including HPV-driven OPSCC, we are happy to be able to address your suggestion by showing classification characteristics related to HPV-driven disease within this table and demonstrate general applicability of the newly developed score. By focusing on the classification characteristics of the leukocyte score (and not stressing differences between HPV-driven cancer vs. other HNSCC) Table 3 contains the desired information about HPV in a way triggering no concerns regarding any overstatement about the influence of HPV on leukocyte subsets.

4)    The first phrase in the Discussion should be erased (it is from the authors guidelines maybe…)

GW: Many thanks for careful reading. You are completely right, and we erased the sentence.

5)    References 14 and 15 are erroneously cited instead of 17&18 in the first paragraph of discussions.

GW: Corrected. 

6)    A discussion about how these technical developments might be applied to other cancers would be very useful. How could this new biomarker proposed for “prognostic classification” and “bias-free detection of HNSCC cancer patients” differentiate between cancers? 

GW: Unfortunately, we do not know anything about differences between cancer entities regarding our leukocyte score as we do not have any other samples as from HNSCC-CP or healthy adults. It might be expected that other cancers behave different and are not linked to the pre-dominant deviating (increased) relative numbers in monocytes, lowered relative proportions of classical (CD14++CD16+), and are therefore equally discriminated applying the score. Otherwise I would rather expect that (given the more intense investigated “big five” cancers) a leukocyte score based on the four ratios would have been published earlier if such an impact on leukocyte subset distribution exists in peripheral blood associated to these cancers. Possibly it depends also on the marker-defined enumeration of leukocyte subsets and use of 10-color FCM for simultaneous analysis…

Regarding the label “bias-free”: We used this label with respect to the type of statistical analyses. Informedness and markedness are the bias- and prevalence-independent alternatives to specificity and sensitivity. Bias is the relative number of positive and negative predictions; prevalence the proportion of (in our case) HNSCC-CP in the total number under investigation. “Bias-free” means independence from the chance to detect positives/give a positive result (the bias was 1:1, 50% in our test cohort, but only 6.2% in our validation cohort). “Prevalence-free” means that the proportion of positives has no effect on the outcome of the comparison (that is reflected in the much nearer results for these measures in the TC, VC and both combined compared with the classical measures sensitivity and specificity which both demonstrate an effect of bias and prevalence of the findings). “Bias-free” does not imply the independence of detection of HNSCC-CP independent from the background, simultaneous present second cancers of other histology or leukemia as “bias-free” addresses only the statistical prerequisites based on the chance to correctly identify cases based on deviating proportions of cases versus controls. According to Powers (2011), “Informedness quantifies how informed a predictor is for the specified condition, and specifies the probability that a prediction is informed in relation to the condition (versus chance); Markedness quantifies how marked a condition is for the specified predictor, and specifies the probability that a condition is marked by the predictor (versus chance).” Informedness = Recall + Inverse Recall – 1 = tpr-fpr (true positive rate – false positive rate), whereas Markedness = Precision + Inverse Precision – 1 = tpa-fna (true positive accuracy [precision] – false negative accuracy). The product (geometric mean) of Informedness and Markedness (Matthew’s Correlation) is a unique prevalence- and bias-independent measure of quality. Matthew’s Correlation above 91% in each of the cohorts of our investigation shows (based on the high true positive rate and accuracy in absence of a relevant false positive rate and false negative accuracy) the usefulness of the leukocyte score independent from the prevalence of HNSCC-CP in a given cohort. Prevalence is regarded as a constant of the target condition or data set (and this constant c is c=[1−Prevalence]/Prevalence), whilst parameterizing or selecting a model can be viewed in terms of trading off the true positive rate (tpr) and the false positive rate (fpr) as in ROC analysis, or equivalently as controlling the relative number of positive and negative predictions, namely the Bias, in order to maximize a particular accuracy measure (Recall, Precision, F-Measure, Rand Accuracy and AUC). Note that for a given Recall level, the other measures (Accuracy and Precision) all decrease with increasing Bias towards positive predictions. That’s why I included the bias- and prevalence independent measures in Figure 4 and stressed the findings in Results and Discussion.

7)    An error appears in the Manual Gating section in Materials and Methods. Plot 1 (defined as Time vs SSC) and plot 2 (FSC TOF vs FSC Peak Lin) are missing from Supplementary Figure 1. They are also different from Plot 1 and 2 in Ref 17. This has to be corrected.

GW: We corrected this by including the so far missing plots in Figure S1 and rendering the former Figure S1 to Figure S2. The respective legends were adapted.

8)    Please correct the Reference list. It appears that the order in the list does not respect the order in the text and some references are not cited.

GW: The reference list was updated/corrected.

 Many thanks for the very helpful comments essentially contributing to a now much better paper!

Reviewer 5 Report

In the present study, Wichmann et al. investigated the prognostic value of cell counts of leukocyte in peripheral blood (PB) in head and neck squamous cell carcinoma cancer patients (HNSCC-CP). They showed that the relative cell counts not necessary the absolute cell counts of leukocyte subsets in PB of HNSCC-CP differ significantly from healthy adults. The topic is interesting. However, the method described is hard to follow. The authors have to address the following my concerns.

1. It is not clear how the relative cell counts of leukocyte subsets in PB are calculated? The definition may have been published in refs 15-16, however, the authors still need to clarify the calculation method in the present study.

2. The definition of the leukocyte score is kind of artificial. The authors have to validate the definition carefully. Maybe the authors can derive and optimize a binary classification model with MGR, MLR, MTLR and clMMR as the features.

3. Without providing cross-validation, it is very difficult to evaluate the performance of their leukocyte score. 

4. Is there any specific reason to use 892 instead of 920 healthy adult samples for analysis in Table 1?

Author Response

However, the method described is hard to follow. The authors have to address the following my concerns.

It is not clear how the relative cell counts of leukocyte subsets in PB are calculated? The definition may have been published in refs 15-16, however, the authors still need to clarify the calculation method in the present study.

GW: There is indeed a statement on that in ref. 16. The total number of cells was analyzed using the routinely used Hematology Analyzer XN-9000 (Sysmex Europe GmbH, Norderstedt, Germany) in the central lab of the University Hospital Leipzig. The colleagues provided the absolute cell counts that were used to calculate the absolute cell count presented in Table 1. Relative cell counts are the number of events within the gate containing those events with the marker combination defining the particular leukocyte subset divided by the total number of leukocytes (the total number of events in the three gates A, B and C in plot 3 in Figure S1 or in plot 1 of Figure S2). As this routinely used procedure to calculate relative cell counts is self-explaining and the required background information is already presented in the legend of Table 2, there is from our point of view no need for further clarification or going more into detail regarding this simple calculation.

The definition of the leukocyte score is kind of artificial. The authors have to validate the definition carefully. Maybe the authors can derive and optimize a binary classification model with MGR, MLR, MTLR and clMMR as the features.

GW: Of course, any mathematical or statistical model is kind of artificial. The definition of cut-off values, however, was driven by the data obtained in the propensity-score matched training cohort using the Youden score, the maximum of sensitivity+specificity obtained for the four ratios MGR, MLR, MTLR and clMMR. I don’t know what further optimization is expected above the high level of accuracy achieved, the prevalence- and bias-independent good performance of the score and the demonstration of its potential value to discriminate HNSCC-CP and HA by only a few misclassified cases. As we had to address the main argument against comparability of HNSCC-CP and HA, that is life-style attributable risk factors for development of HNSCC-CP (in particular high level of daily alcohol consumption, smoking and a history of many pack years cigarettes smoked), we used propensity-score matching to derive cut-offs for MGR, MLR, MTLR and clMMR from the matched HNSCC-CP and HA. The ROC curves for MGR, MLR, MTLR and clMMR provided good cut-off points for binary classification also of the VC, and a further optimization using binary classification by using weighted ratios would probably lead to overfitting and/or require the use of more parameters. Moreover, making things high sophisticated is the best way to prevent their use. Otherwise it might be useful to develop and optimize a binary classification after performing a prospective validation study combining both datasets.

Without providing cross-validation, it is very difficult to evaluate the performance of their leukocyte score. 

GW: We performed cross-validation applying stratified bootstrapping utilizing the bias-corrected accelerated method and achieved significant values for Lambda (0.860, BCa 95% CI 0.780 – 0.928; p <10-18), Goodman & Kruskal Tau (0.851, BCa 95% CI 0.772 – 0.923; p=1.19 ∙10-198) and the uncertainty coefficient (0.798, BCa 95% CI 0.711 – 0.885; p=3.81∙10-117). Cramer-V (0.923, BCa 95% CI 0.879 – 0.961), Phi (0.923, BCa 95% CI 0.879 – 0.961) and the contingency coefficient (C=0.678, BCa 95% CI 0.660 – 0.693; p=7.74 ∙10-199) demonstrate that the null hypothesis H0 can be rejected without high level of uncertainty (OR predicted HA/HNSCC-CP 1814.4, BCa 95% CI 605.5 – 12085.4, within HA 14.18, BCa 95% CI 7.64 – 47.7, and within HNSCC-CP 0.008, BCa 95% CI 0.003 – 0.013). According to Mantel-Haensel estimates we found an OR of 1814.4 (95% CI 623.0 – 5284.8; p=4.65 ∙10-43) under the assumption of a combined ratio of 1; using the bootstrap applying 1,000 iterations for estimating the confidence intervals for the natural logarithm of the Mantel-Haensel estimate we detected an estimated ln of 7.504 (Mantel-Haensel BCa 95% CI 6.445 – 9.377; p=9.99 ∙10-4). We detected a type I error (Fallout) of α=0.007 and a type II error (Miss rate) of β=0.07. Also Bayesian statistics indicated the high quality of predictions applying the leukocyte score as the probability being a HNSCC-CP if a leukocyte score ≥3 is detected P (HNSCC-CP│leukocyte score ≥3) was found to be 100%, 88.5% and 93.0% in TC, VC and in both cohorts combined. However, it seems to be a little bit too much to present all these data in a paper addressing the use of OMIP-023 and simultaneous detection of 15 leukocyte subsets in peripheral blood of 101 HNSCC-CP and 963 HA and the use of 4 ratios of marker-defined leukocyte subsets for detection of HNSCC-CP. We therefore added only the following sentences into 2.3. Development… :

Bootstrapping utilizing the bias-corrected accelerated (BCa) method revealed significant values for Lambda (0.860, BCa 95% CI 0.780 – 0.928; p <10-18), Goodman & Kruskal Tau (0.851, BCa 95% CI 0.772 – 0.923; p=1.19 ∙10-198) and the uncertainty coefficient (0.798, BCa 95% CI 0.711 – 0.885; p=3.81∙10-117). Cramer-V (0.923, BCa 95% CI 0.879 – 0.961), Phi (0.923, BCa 95% CI 0.879 – 0.961) and the contingency coefficient C (0.678, BCa 95% CI 0.660 – 0.693; p=7.74 ∙10-199) equally demonstrate a good performance of the score. Cross validation within the Mantel-Haenszel statistics to estimate conditional probability revealed significance for the combined odds ratio (3254.9, BCa 95% CI 613.1 – 17280.0; p=2.19∙10-21) even on the natural logarithm scale (ln OR=8.088, BCa 95% CI 6.797 – 10.030; p=1.06∙10-3). The a-posteriori (Bayes) probability P (HNSCC-CP│leukocyte score ≥3) was found to be 100%, 88.5% and 93.0% in TC, VC and in both cohorts combined.

… and the following sentence into Discussion:

On top of that, Bayesian statistics indicated the high a-posteriori probability being a HNSCC-CP whenever the leukocyte score is ≥3: P (HNSCC-CP│leukocyte score ≥3) in TC, VC and in both cohorts combined are 100%, 88.5% and 93.0%, respectively.

We updated also the M&M section by adding the following sentence into 4.7. Statistical Analyses:

Cross validation was executed using SPSS version 24 (IBM Corporation, Armonk, NY, USA) applying bootstrapping of 1,000 iterations and the bias-corrected accelerated (BCa) method to validate the classification characteristics of the leukocyte score.

4. Is there any specific reason to use 892 instead of 920 healthy adult samples for analysis in Table 1?

GW: As stated above, the total number of cells was analyzed using the routinely used Hematology Analyzer XN-9000 (Sysmex Europe GmbH, Norderstedt, Germany) in the central lab of the University Hospital Leipzig. The colleagues provided the absolute cell counts that were used to calculate the absolute cell count presented in Table 1. Unfortunately, this was not the case for those 28 missing samples which could therefore not be included in the comparisons presented in Table 1. The 10-color FCM using the OMIP-023 was performed on a Navios flow cytometer that is not able to provide absolute cell counts but an ideal tool to perform cytomics based on relative cell counts. However, the relative cell counts were available for the specified number of HNSCC-CP and HA in the training cohort (TC), validation cohort (VC), and both combined (Table 2).

Round 2

Reviewer 1 Report

This is a revised version. Although the paper improved in some aspects I am still not convinced that the scores are sensitive and specific enough to detect HNSCC.

Author Response

Comments and Suggestions for Authors

This is a revised version. Although the paper improved in some aspects I am still not convinced that the scores are sensitive and specific enough to detect HNSCC.

GW: Many thanks for the second review of our paper. There is a statement that you are “still not convinced that the scores are sensitive and specific enough to detect HNSCC”. Unfortunately, I cannot understand the difficulties to accept that the high sensitivity and specificity, but more importantly the accuracy of 95.3%, 99.0% and 98.7% in the Propensity-score matched training cohort (TC), the validation cohort (VC) and both cohorts combined (BCC), and the height of the bias- and prevalence-independent measures informedness (90.7%, 94.0% and 92.3%), markedness (91.5%, 88.2% and 92.3%) and Matthew’s correlation of 91.1%, 91.0% and 92.3% together with the odds ratios (corrected odds ratios) all with the lower limits of their 95% confidence intervals above 200 clearly indicate the ability of the score to detect HNSCC among a population-based cohort with high confidence. If you are one of the frequentists rather accepting p values than confidence intervals regarding the acceptation or rejection of the alternative hypothesis, p values of 2.97 ∙ 10-17, 4.04 ∙ 10-178, and 7.74 ∙ 10-199, in TC, VC and BCC may convince you that the score’s classification characteristics demonstrated by various measures warrant further research. Additionally performed cross-validation applying stratified bootstrapping utilizing the bias-corrected accelerated method revealed significant values for Lambda (0.860, BCa 95% CI 0.780 – 0.928; p <10-18), Goodman & Kruskal Tau (0.851, BCa 95% CI 0.772 – 0.923; p=1.19 ∙10-198) and the uncertainty coefficient (0.798, BCa 95% CI 0.711 – 0.885; p=3.81∙10-117). Cramer-V (0.923, BCa 95% CI 0.879 – 0.961), Phi (0.923, BCa 95% CI 0.879 – 0.961) and the contingency coefficient (C=0.678, BCa 95% CI 0.660 – 0.693; p=7.74 ∙10-199) demonstrate that the null hypothesis H0 can be rejected without high level of uncertainty (OR predicted HA/HNSCC-CP 1814.4, BCa 95% CI 605.5 – 12085.4, within HA 14.18, BCa 95% CI 7.64 – 47.7, and within HNSCC-CP 0.008, BCa 95% CI 0.003 – 0.013). According to Mantel-Haensel estimates we found an OR of 1814.4 (95% CI 623.0 – 5284.8; p=4.65 ∙10-43)  under the assumption of combined ratio of 1; using the bootstrap applying 1,000 iterations for estimating the confidence intervals for the natural logarithm of the Mantel-Haensel estimate we detected an estimated ln of 7.504 (Mantel-Haensel BCa 95% CI 6.445 – 9.377; p=9.99 ∙10-4). Cross validation within the Mantel-Haenszel statistics to estimate conditional probability revealed significance for the combined odds ratio (3254.9, BCa 95% CI 613.1 – 17280.0; p=2.19∙10-21) even on the natural logarithm scale (ln OR=8.088, BCa 95% CI 6.797 – 10.030; p=1.06∙10-3).  We detected a low type I error (Fallout= α=0.007) and also the type II error (Miss rate= β=0.07) appears to be acceptable. Also Bayesian statistics indicated the high quality of predictions applying the leukocyte score ≥3 as criterion as the probability being a HNSCC-CP if a leukocyte score ≥3 is detected P (HNSCC-CP│leukocyte score ≥3) was found to be 100%, 88.5% and 93.0% in TC, VC and in both cohorts combined.

Therefore, the use of the OMIP-023 panel and 10-color FCM may allow for detection of HNSCC among healthy adults of a population-based cohort as suggested by the significant findings independent from the type of statistics applied. That is what our paper says. Of course, a prospective validation of our findings in a follow-up study by an independent group may definitely clarify your doubts and concerns. But this, however, can be done only after publication of our results. So please judge about the scientific quality of the paper and the provided statistics therein and outlined here, indicate your name on the review and we may be able to perform the prospective investigations together.

Reviewer 5 Report

The authors have appropriately addressed my concerns.